# Learning to Time-Decode in Spiking Neural Networks Through the Information Bottleneck

**Nicolas Skatchkovsky**
KCLIP lab, Dept. of Eng.
King's College London
London, UK
nicolas.skatchkovsky@kcl.ac.uk

**Osvaldo Simeone**
KCLIP lab, Dept. of Eng.
King's College London
London, UK
osvaldo.simeone@kcl.ac.uk

**Hyeryung Jang**
ION group, Dept. of A.I.
Dongguk University
Seoul, Korea
hyeryung.jang@dgu.ac.kr

## Abstract

One of the key challenges in training Spiking Neural Networks (SNNs) is that target outputs typically come in the form of natural signals, such as labels for classification or images for generative models, and need to be encoded into spikes. This is done by handcrafting target spiking signals, which in turn implicitly fixes the mechanisms used to decode spikes into natural signals, e.g., rate decoding. The arbitrary choice of target signals and decoding rule generally impairs the capacity of the SNN to encode and process information in the timing of spikes. To address this problem, this work introduces a hybrid variational autoencoder architecture, consisting of an encoding SNN and a decoding Artificial Neural Network (ANN). The role of the decoding ANN is to learn how to best convert the spiking signals output by the SNN into the target natural signal. A novel end-to-end learning rule is introduced that optimizes a directed information bottleneck training criterion via surrogate gradients. We demonstrate the applicability of the technique in an experimental settings on various tasks, including real-life datasets.

## 1 Introduction

While traditional Artificial Neural Networks (ANNs) implement static rate-based neurons, Spiking Neural Networks (SNNs) incorporate dynamic spike-based neuronal models that are closer to their biological counterparts. In different communities, SNNs are used either as models for the operation of biological brains or as efficient solutions for cognitive tasks. The efficiency of SNNs, when implemented on dedicated hardware, hinges on the high capacity, low latency, and high noise tolerance of information encoding in the timing of spikes [1]. Recent results leveraging novel prototype chips confirm the potential order-of-magnitude gains in terms of latency- and energy-to-accuracy metrics for several optimization and inference problems [2]. In contrast, the landscape of training algorithms for SNNs is still quite fractured [2], and a key open question is how to derive learning rules that can optimally leverage the time encoding capabilities of spiking neurons. This is the focus of this work.

**Learning how to decode.** Consider the problem of training an SNN – a network of spiking neurons. To fix the ideas, as illustrated in Fig. 1(a)-(b), say that we wish to train an SNN to classify inputs from a neuromorphic sensor, such as a DVS camera [5], which are in the form of a collection of spiking signals. We hence have a spiking signal as input and a natural signal (the label) as output. As another example in Fig. 1(e), we may wish to "naturalize" the output of a DVS camera to produce the corresponding natural image [6]. The direct training of an SNN hinges on the definition of a loss function operating on the output of the SNN. This is typically done by converting the target natural signals into spiking signals that serve as the desired output of the SNN. As seen in Fig. 1(a)-(b), one could for instance assign a large-rate spiking signal or an earlier spike to the readout neuron representing the correct class. The loss function then measures the discrepancy between actual output spiking signals and the corresponding spiking targets.

35th Conference on Neural Information Processing Systems (NeurIPS 2021).

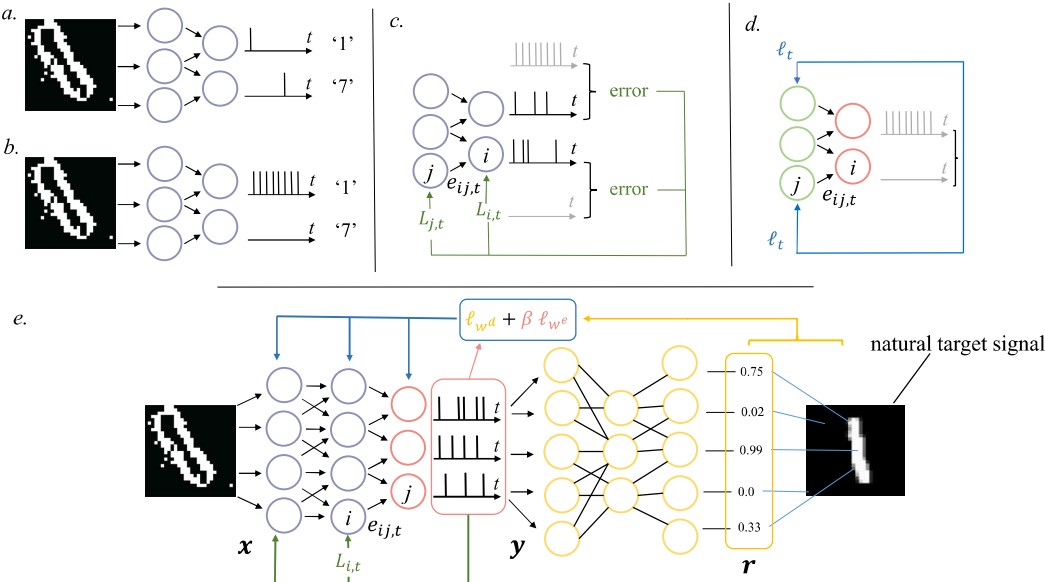

Figure 1: (a) Illustration of an SNN trained for image classification with first-to-spike decoding. (b) Illustration of an SNN trained for image classification with rate decoding. (c) Illustration of an SG scheme [3]: At each time-step, updates for neuron $i$ are computed using local eligibility traces $\boldsymbol{e}_{i,t}$, and per-neuron learning signals $L_{i,t}$, obtained from errors computed based on handcrafted target signals (gray). (d) Illustration of ML learning for SRM with stochastic threshold [4]: Updates for *readout* neurons (green) are obtained with local eligibility traces based on handcrafted target spiking signal (gray), while updates for *hidden* neurons (red) make use of a common learning signal $\ell_t$ based on the same target signal. (e) Illustration of the proposed hybrid variational autoencoder: An encoding SNN processes inputs from a spiking image source, such as a DVS camera, to produce a spiking latent representation. A decoding ANN learns to decode this representation with the aim of reconstructing a natural target signal, such as an image, corresponding to the spiking input of the SNN. SNN and ANN are jointly trained under a directed information bottleneck criterion. Accordingly, training of the ANN is based on backpropagation, while training of the SNN uses per-neuron learning signals and common feedback from both networks.

The approach outlined in the previous paragraph has an important problem: By selecting handcrafted target spiking signals, one is effectively choosing a priori the type of information encoding into spikes that the SNN is allowed to produce. This constrains the capacity of the SNN to fully leverage the time encoding capabilities of spiking signals. For instance, the target signals outlined in Fig. 1(a) correspond to assuming first-to-spike decoding, while those in Fig. 1(b) to rate decoding [7]. This paper proposes a method that allows the SNN to automatically learn how to best encode the input into spiking signals, without constraining a priori the decoding strategy.

To this end, we introduce the **hybrid SNN-ANN autoencoder** architecture shown in Fig. 1(e) that consists of an encoding SNN to be trained alongside a decoding ANN. The role of the decoding ANN is to learn how to best convert the spiking signals output by the SNN into the target natural signal. Encoding SNN and decoding ANN are trained end-to-end so as to ensure that the SNN produces informative spike-based representations that can be efficiently decoded into the desired natural signals.

**Directed information bottleneck.** How to jointly train the hybrid SNN-ANN autoencoder? On the one hand, we wish the spiking output of the SNN to be informative about the target natural signal – which we will refer to as *reference signal* – and, on the other, we would like the spiking output of the SNN to remove all extraneous information present in the spiking input that is not informative about the reference signal. To capture this dual requirement, we adopt an *information bottleneck* (IB) formulation [8], and view the ANN as the inference network in a *variational autoencoder* system [9].

In applying the *variational IB* methodology to the hybrid SNN-ANN autoencoder in Fig. 1(e), we make two key contributions. 1) Given the inherently dynamic operation of the SNN, we generalize the

IB criterion to apply to directed information metrics [10], in lieu of conventional mutual information. We refer to the resulting approach as **variational directed IB (VDIB)**. 2) In order to enable the optimization of information-theoretic metrics, such as the IB, we introduce an SNN model that encompasses *deterministic hidden neurons and probabilistic readout neurons*. In so doing, we combine two strands of research on training SNNs: The first line focuses purely on deterministic models, viewing an SNN as a Recurrent Neural Network (RNN) and deriving approximations of backpropagation-through-time (BPTT) through surrogate gradient (SG) techniques [3, 11–15]; while the second assumes probabilistic neurons only, deriving approximations of maximum likelihood (ML) learning. Additional discussion on related work can be found in the next section.

## 2    Related works

**Training SNNs.**    In recent years, SG techniques for training SNNs have gained significant momentum, becoming the state-of-the-art and *de-facto* eclipsing other techniques. As reviewed in [12], these techniques rely on surrogate derivatives to replace the ill-defined derivative of the step activation function of spiking neurons. The SGs are combined with truncated and approximated versions of BPTT, including the use of per-layer local error signals [13] and per-neuron learning signals defined using direct feedback alignment [3, 15]. Learning rules for probabilistic SNNs are typically derived by tackling the ML problem, and result in updates that rely on random sampling and common – rather than per-neuron – learning signals. The approach was shown to generalize to networks of winner-take-all neurons [16]. As illustrated in Fig. 1, the proposed approach builds on both models to enable the use of SG methods for the optimization of probabilistic learning criterion.

**Hybrid SNN-ANN architectures.**    A hybrid SNN-ANN architecture has been recently proposed in [17] for gesture similarity analysis, based on the variational autoencoder (VAE) framework. Although yielding a similar objective, the encoder in [17] is not fully spiking since the ANN receives the membrane potentials of the spiking neurons in the readout layer of the SNN. The model proposed in our work circumvents this limitation.

**Information bottleneck.**    The IB technique originated two decades ago [8] in the context of a rate-distortion problem formation assuming discrete variables. The approach has regained interest in recent years, first as a means to explain the performance of ANNs [18], and then as a training criterion in [9]. We extend the IB by considering temporal signals under causality constraints, hereby replacing the mutual information with directed information [10]. The IB principle has been previously used in the SNN literature in the case of a single neuron [19, 20]. Finally, it has been used in theoretical neuroscience to explain processing in sensory neurons [21] and as a unifying principle for the predictive and efficient coding theories. Our model encompasses the framework introduced in [21].

## 3    Problem definition

**Notations.**    Consider three jointly distributed random vectors $x, y$, and $z$ with distribution $p(x, y, z)$. The conditional mutual information (MI) between $x$ and $y$ given $z$ is defined as $I(x; y|z) = \mathrm{E}_{p(x,y,z)}\left[\log\left(\frac{p(x,y|z)}{p(x|z)p(y|z)}\right)\right]$, where $\mathrm{E}_p[\cdot]$ represents the expectation with respect to the distribution $p$. The Kullback-Leibler (KL) divergence between two probability distributions $p(x)$ and $q(x)$ defined on the same probability space is $\mathrm{KL}(p(x)||q(x)) = \mathrm{E}_{p(x)}\left[\log\left(\frac{p(x)}{q(x)}\right)\right]$.

For any two jointly distributed random vectors $a = (a_1, \dots, a_T)$ and $b = (b_1, \dots, b_T)$ from time 1 to $T$, the $\tau$-causally conditioned distribution of $a$ given $b$ is defined as

$$p^{(\tau)}(a||b) = \prod_{t=1}^{T} p(a_t|a^{t-1}, b_{t-\tau}^t), \qquad (1)$$

where we denote as $a_{t_1}^{t_2} = (a_{t_1+1}, \dots, a_{t_2})$ the overall random vector from time $t_1 + 1$ to $t_2$, and write $a^t = a_1^t$; This distribution (1) captures the causal dependence of sequence $a$ on the last $\tau$ samples of $b$. If $\tau = T$, we recover the causally conditioned distribution [10].

We also define the $\tau$-directed information with $\tau \geq 0$ as

$$I^{(\tau)}(a \to b) = \sum_{t=1}^{T} I(a_{t-\tau}^t; b_t|b^{t-1}). \qquad (2)$$

This metric quantifies the causal statistical dependence of the last $\tau$ samples from sequence $\boldsymbol{a}$ to $\boldsymbol{b}$. With $\tau = T$, this recovers the standard directed information [10]. We denote as $*$ the convolution operator $f_t * g_t = \sum_{\delta \geq 0} f_\delta g_{t-\delta}$.

**Problem definition.** We study the hybrid SNN-ANN autoencoder architecture in Fig. 1 with the aim of training an SNN encoder to produce a time-encoded spiking representation $\boldsymbol{y}$ of the spiking input $\boldsymbol{x}$ that is informative about a natural reference signal $\boldsymbol{r}$. Throughout, a *spiking signal* is a binary vector sequence, with "1" representing a spike, and a *natural signal* is a real-valued vector sequence. In the prototypical application in Fig. 1(e), the input $\boldsymbol{x}$ is obtained from a neuromorphic sensor, such as a DVS camera [5], and the reference signal $\boldsymbol{r}$ represents the corresponding target natural signal, such as a label or the original natural image. The spiking representation $\boldsymbol{y}$ produced by the SNN is fed to an ANN, whose role is to decode $\boldsymbol{y}$ into the reference signal $\boldsymbol{r}$. Importantly, we do not *a priori* constrain the decoding strategy. In contrast, existing works typically handcraft target spiking signals $\boldsymbol{y}$ corresponding to some fixed decoding rule, such as rate decoding (Fig. 1(a)-(b)). Our main contribution is a method to train encoding SNN and decoding ANN jointly, with the goal of discovering automatically efficient time-decoding rules from spiking signals produced by the encoding SNN.

To optimize SNN encoder and ANN decoder, we consider a data set comprising $N$ pairs $(\boldsymbol{x}, \boldsymbol{r})$ of exogeneous spiking inputs $\boldsymbol{x}$ and natural reference signals $\boldsymbol{r}$ generated from an unknown population distribution $p(\boldsymbol{x}, \boldsymbol{r})$. The exogeneous inputs are in the form of arbitrary time vector sequence $\boldsymbol{x} = (\boldsymbol{x}_1, \ldots, \boldsymbol{x}_T)$ with $\boldsymbol{x}_t \in \{0, 1\}^{N_X}$, while the reference signals $\boldsymbol{r} = (\boldsymbol{r}_1, \ldots, \boldsymbol{r}_T)$, with $\boldsymbol{r}_t \in \mathbb{R}^{N_R}$, define general target signals. In the image naturalization task illustrated in Fig. 1, the exogenous inputs $\boldsymbol{x}$ are the spiking signals encoding the input from a DVS camera, with $N_X$ being the number of pixels; and the reference signals $\boldsymbol{r}$ encode the pixel intensities of the natural image. This can be done for instant by setting $\boldsymbol{r}_t = \boldsymbol{0}_{N_R}$ for $t = 1, \ldots, T - 1$, and $\boldsymbol{r}_T$ equal to the grayscale image of $N_R$ pixels; or setting $\boldsymbol{r}_t$ to equal the grayscale image of all $t = 1, \ldots, T$.

We wish to train the encoding SNN to output a spiking representation $\boldsymbol{y}$ that is maximally informative about the natural target signal $\boldsymbol{r}$. As we will detail below, the SNN implements a causal stochastic mapping between latent signal $\boldsymbol{y}$ and input $\boldsymbol{x}$ that depends on a vector of parameters $\boldsymbol{w}^e$ as

$$p_{\boldsymbol{w}^e}^{(\tau_e)}(\boldsymbol{y}||\boldsymbol{x}) = \prod_{t=1}^T p(\boldsymbol{y}_t|\boldsymbol{y}^{t-1}, \boldsymbol{x}_{t-\tau_e}^t), \tag{3}$$

where the integer $\tau_e$ represents the memory of the encoding SNN and $\boldsymbol{w}^e$ the synaptic weights of the SNN. As a starting point, consider maximizing the $\tau_d$-directed information between $\boldsymbol{y}$ and $\boldsymbol{r}$ as

$$\max_{\boldsymbol{w}^e} I_{\boldsymbol{w}^e}^{(\tau_d)}(\boldsymbol{y} \to \boldsymbol{r}), \tag{4}$$

for a decoding window $\tau_d \geq 0$, over the vector $\boldsymbol{w}^e$ of the encoding SNN parameters. The use of the $\tau_d$-directed information, in lieu of the mutual information $I(\boldsymbol{y}; \boldsymbol{r})$, reflects the practical requirement that reference signal $\boldsymbol{r}_t$ be well represented by a limited window $\boldsymbol{y}_{t-\tau_d}^t$ of past outputs of the SNN in a causal manner. The directed information $I_{\boldsymbol{w}^e}^{(\tau_d)}(\boldsymbol{y} \to \boldsymbol{r})$ is evaluated with respect to the marginal $p(\boldsymbol{y}, \boldsymbol{r})$ of the joint distribution

$$p_{\boldsymbol{w}^e}(\boldsymbol{x}, \boldsymbol{y}, \boldsymbol{r}) = p(\boldsymbol{x}, \boldsymbol{r}) p_{\boldsymbol{w}^e}^{(\tau_e)}(\boldsymbol{y}||\boldsymbol{x}), \tag{5}$$

combining population distribution and SNN mapping via the chain rule of probability.

The maximization (4) does not impose any constraint on the complexity of the spiking representation $\boldsymbol{y}$. To address this problem, we adopt the **directed information bottleneck (DIB)** objective

$$R_{\text{DIB}}(\boldsymbol{w}^e) = I_{\boldsymbol{w}^e}^{(\tau_d)}(\boldsymbol{y} \to \boldsymbol{r}) - \beta \cdot I_{\boldsymbol{w}^e}^{(\tau_e)}(\boldsymbol{x} \to \boldsymbol{y}), \tag{6}$$

where $\beta > 0$ is a hyperparameter. The penalty term $I_{\boldsymbol{w}^e}^{(\tau_e)}(\boldsymbol{x} \to \boldsymbol{y})$ is the directed information between the input $\boldsymbol{x}$ and the spiking representation $\boldsymbol{y}$, which captures the complexity of the causal encoding done by the SNN. This term is evaluated with respect to the marginal $p(\boldsymbol{x}, \boldsymbol{y}) = p(\boldsymbol{x}) p_{\boldsymbol{w}^e}^{(\tau_e)}(\boldsymbol{y}||\boldsymbol{x})$ of the joint distribution (5), and it reflects the limited memory of the SNN encoding mapping (3). It can be interpreted as the amount of information about input $\boldsymbol{x}$ that is (causally) preserved by the representation $\boldsymbol{y}$. To the best of our knowledge, the DIB objective is considered in this paper for the first time.

# 4 Learning to decode through the IB

In order to address the DIB problem (6), we adopt a variational formulation that relies on the introduction of a decoding network, as in the architecture illustrated in Fig. 1(d). The decoding network implements the causally conditional distribution

$$q_{\boldsymbol{w}^d}^{(\tau_d)}(\boldsymbol{r}||\boldsymbol{y}) = \prod_{t=1}^{T} q_{\boldsymbol{w}^d}(\boldsymbol{r}_t|\boldsymbol{r}^{t-1}, \boldsymbol{y}_{t-\tau_d}^t) \tag{7}$$

between spiking representation $\boldsymbol{y}$ and natural reference signal $\boldsymbol{r}$. The decoder (7) is causal, with memory given by integer $\tau_d > 1$ and is parametrized by a vector $\boldsymbol{w}^d$. The decoder can be implemented in several ways. A simple approach, that we follow in the remainder of this paper, is to use a model $q_{\boldsymbol{w}^d}(\boldsymbol{r}_t|\boldsymbol{r}^{t-1}, \boldsymbol{y}_{t-\tau_d}^t) = q_{\boldsymbol{w}^d}(\boldsymbol{r}_t|\boldsymbol{y}_{t-\tau_d}^t)$, where $q_{\boldsymbol{w}^d}(\boldsymbol{r}_t|\boldsymbol{y}_{t-\tau_d}^t)$ is an ANN with inputs given by a window $\boldsymbol{y}_{t-\tau_d}^t$ of $\tau_d$ samples from the spiking representation $\boldsymbol{y}$. Alternatively, one could use an RNN to directly model the kernel $q_{\boldsymbol{w}^d}(\boldsymbol{r}_t|\boldsymbol{r}^{t-1}, \boldsymbol{y}_{t-\tau_d}^t)$. Note that the introduction of the decoder network (7) is consistent with the use of the $\tau_d$-directed information in (6). We emphasize that the use of an ANN, or RNN, for decoding, is essential in order to allow the reference sequence $\boldsymbol{r}$ to be a natural signal.

With such a network, using a standard variational inequality [9], we bound the two terms in (6) to obtain the **variational DIB (VDIB)** loss $\mathcal{L}_{\text{VDIB}}(\boldsymbol{w}^e, \boldsymbol{w}^d)$ as an upper bound on the negative DIB objective (6)

$$\mathcal{L}_{\text{VDIB}}(\boldsymbol{w}^e, \boldsymbol{w}^d) = \underbrace{\mathrm{E}_{p_{\boldsymbol{w}^e}(\boldsymbol{y}, \boldsymbol{r})}\left[ -\log q_{\boldsymbol{w}^d}^{(\tau_d)}(\boldsymbol{r}||\boldsymbol{y}) \right]}_{\text{average decoder's log-loss}} + \beta \cdot \mathrm{E}_{p(\boldsymbol{x})}\left[ \underbrace{\mathrm{KL}(p_{\boldsymbol{w}^e}^{(\tau_e)}(\boldsymbol{y}||\boldsymbol{x})||q(\boldsymbol{y}))}_{\text{information theoretic regularization}} \right], \tag{8}$$

in which we have introduced an arbitrary "prior" distribution $q(\boldsymbol{y}) = \prod_t q(\boldsymbol{y}_t|\boldsymbol{y}^{t-1})$ on the spiking representation. As in the implementation in [9] for the standard IB criterion, we will consider $q(\boldsymbol{y})$ to be fixed, although it can potentially be optimized on. To enforce sparsity of the encoder's outputs, the "prior" $q(\boldsymbol{y})$ can be chosen as a sparse Bernoulli distribution [4], i.e., $q(\boldsymbol{y}) = \prod_{t=1}^{T} \text{Bern}(\boldsymbol{y}_t|p)$ for some small probability $p$. The derivation of the bound can be found in Appendix A.1.

The learning criterion (8) enables the joint training of encoding SNN and decoding network via the problem

$$\min_{\boldsymbol{w}^e, \boldsymbol{w}^d} \mathcal{L}_{\text{VDIB}}(\boldsymbol{w}^e, \boldsymbol{w}^d), \tag{9}$$

which we address via SGD. To this end, we now give general expressions for the gradients of the VDIB criterion (8) with respect to the encoder and decoder weights by leveraging the REINFORCE Monte Carlo gradients. In the next section, we will then detail how the gradient with respect to the encoding SNN weights $\boldsymbol{w}^e$ can be approximated by integrating SG [3] with the probabilistic approach [4]. To start, the VDIB loss in (8) can equivalently be stated as

$$\mathcal{L}_{\text{VDIB}}(\boldsymbol{w}^e, \boldsymbol{w}^d) = \mathrm{E}_{\underbrace{p(\boldsymbol{x}, \boldsymbol{r})}_{\text{population distribution}}} \mathrm{E}_{\underbrace{p_{\boldsymbol{w}^e}^{(\tau_e)}(\boldsymbol{y}||\boldsymbol{x})}_{\text{enc. SNN}}} \left[ \underbrace{-\log q_{\boldsymbol{w}^d}^{(\tau_d)}(\boldsymbol{r}||\boldsymbol{y})}_{:= \ell_{\boldsymbol{w}^d}(\boldsymbol{y}, \boldsymbol{r})} + \beta \cdot \underbrace{\log\left( \frac{p_{\boldsymbol{w}^e}^{(\tau_e)}(\boldsymbol{y}||\boldsymbol{x})}{q(\boldsymbol{y})} \right)}_{:= \ell_{\boldsymbol{w}^e}(\boldsymbol{y}, \boldsymbol{x})} \right], \tag{10}$$

where we have defined the decoder loss $\ell_{\boldsymbol{w}^d}(\boldsymbol{y}, \boldsymbol{r})$ and the encoder loss $\ell_{\boldsymbol{w}^e}(\boldsymbol{y}, \boldsymbol{x})$, as shown in (10). The gradient with respect to the encoder weights $\boldsymbol{w}^e$ can be obtained via the REINFORCE gradient technique [22, Ch. 8] [23] as

$$\nabla_{\boldsymbol{w}^e}\mathcal{L}_{\text{VDIB}}(\boldsymbol{w}^e, \boldsymbol{w}^d) = \mathrm{E}_{p(\boldsymbol{x}, \boldsymbol{r})}\mathrm{E}_{p_{\boldsymbol{w}^e}^{(\tau_e)}(\boldsymbol{y}||\boldsymbol{x})}\left[ \left( \ell_{\boldsymbol{w}^d}(\boldsymbol{y}, \boldsymbol{r}) + \beta \cdot \ell_{\boldsymbol{w}^e}(\boldsymbol{y}, \boldsymbol{x}) \right) \cdot \nabla_{\boldsymbol{w}^e}\log p_{\boldsymbol{w}^e}^{(\tau_e)}(\boldsymbol{y}||\boldsymbol{x}) \right], \tag{11}$$

while the gradient with respect to the decoder weights $\boldsymbol{w}^d$ can be directly computed as

$$\nabla_{\boldsymbol{w}^d}\mathcal{L}_{\text{VDIB}}(\boldsymbol{w}^e, \boldsymbol{w}^d) = \mathrm{E}_{p(\boldsymbol{x}, \boldsymbol{r})}\mathrm{E}_{p_{\boldsymbol{w}^e}^{(\tau_e)}(\boldsymbol{y}||\boldsymbol{x})}\left[ \nabla_{\boldsymbol{w}^d}\ell_{\boldsymbol{w}^d}(\boldsymbol{y}, \boldsymbol{r}) \right]. \tag{12}$$

Unbiased estimates of the gradients (11)-(12) can be evaluated by using one sample $(\boldsymbol{x}, \boldsymbol{r}) \sim p(\boldsymbol{x}, \boldsymbol{r})$ from the data set, along with a randomly generated SNN output $\boldsymbol{y} \sim p_{\boldsymbol{w}^e}^{(\tau_e)}(\boldsymbol{y}||\boldsymbol{x})$. We obtain

$$\nabla_{\boldsymbol{w}^e} \mathcal{L}_{\text{VDIB}}(\boldsymbol{w}^e, \boldsymbol{w}^d) \approx \Big(\ell_{\boldsymbol{w}^d}(\boldsymbol{y}, \boldsymbol{r}) + \beta \cdot \ell_{\boldsymbol{w}^e}(\boldsymbol{y}, \boldsymbol{x})\Big) \cdot \nabla_{\boldsymbol{w}^e} \log p_{\boldsymbol{w}^e}^{(\tau_e)}(\boldsymbol{y}||\boldsymbol{x}) \tag{13}$$

$$\nabla_{\boldsymbol{w}^d} \mathcal{L}_{\text{VDIB}}(\boldsymbol{w}^e, \boldsymbol{w}^d) \approx \nabla_{\boldsymbol{w}^d} \ell_{\boldsymbol{w}^d}(\boldsymbol{y}, \boldsymbol{r}). \tag{14}$$

While the gradient $\nabla_{\boldsymbol{w}^d} \ell_{\boldsymbol{w}^d}(\boldsymbol{y}, \boldsymbol{r})$ can be computed using standard backpropagation on the decoding ANN, the gradient $\nabla_{\boldsymbol{w}^e} \log p_{\boldsymbol{w}^e}^{(\tau_e)}(\boldsymbol{y}||\boldsymbol{x})$ for the encoder weights depends on the SNN model, and is detailed in the next section.

As we will describe in the next section, and as illustrated in Fig. 1(d), the learning rule for the SNN involves the common learning signal $\big(\ell_{\boldsymbol{w}^d}(\boldsymbol{y}, \boldsymbol{r}) + \beta \cdot \ell_{\boldsymbol{w}^e}(\boldsymbol{y}, \boldsymbol{x})\big)$ which includes two feedback signals, one from the ANN decoder — the loss $\ell_{\boldsymbol{w}^d}(\boldsymbol{y}, \boldsymbol{r})$ — and one from the SNN encoder, namely $\ell_{\boldsymbol{w}^e}(\boldsymbol{y}, \boldsymbol{x})$. The latter imposes a penalty for the distribution of the representation $\boldsymbol{y}$ that depends on its "distance", in terms of log-likelihood ratio, with respect to the reference distribution $q(\boldsymbol{y})$. Apart from the described common learning signals, which is a hallmark of probabilistic learning rules [4, 24], the update also involves per-neuron terms derived as in SG-based rules. The proposed algorithm is detailed in Appendix A.1 and derived next.

## 5 SNN Model

The encoding SNN is a directed network $\mathcal{N}$ of spiking neurons for which at each time-step $t = 1, \ldots, T$, each neuron $i \in \mathcal{N}$ outputs a binary value $z_{i,t} \in \{0, 1\}$. The set of neurons can be partitioned as $\mathcal{N} = \mathcal{H} \cup \mathcal{Y}$, where $\mathcal{H}$ denotes the set of $N_H$ *hidden* neurons, and $\mathcal{Y}$ the set of $N_Y$ *readout* neurons that output spiking signal $\boldsymbol{y}$ over $T$ time instants. The SNN defines the stochastic mapping $p_{\boldsymbol{w}^e}^{(\tau_e)}(\boldsymbol{y}||\boldsymbol{x})$ in (3) between exogenous input sequence $\boldsymbol{x}$ and the output $\boldsymbol{y}$ of the readout neurons. For each neuron $i \in \mathcal{N}$, we denote as $\mathcal{P}_i$ the set of *pre-synaptic* neurons, i.e., the set of neurons having directed synaptic links to neuron $i$. With a slight abuse of notation, we include the $N_X$ exogenous inputs in this set. In order to enable the use of information-theoretic criteria via the stochastic mapping (3), as well as the efficiency of SG-based training, we model the readout neurons in $\mathcal{Y}$ using the (probabilistic) Spike Response Model (SRM) [25] with stochastic threshold [4], while the hidden neurons in $\mathcal{H}$ follow the standard (deterministic) SRM. Accordingly, the spiking mechanism of every neuron $i \in \mathcal{N}$ in the SNN depends on the spiking history of pre-synaptic neurons and of the post-synaptic neuron itself through the neuron's *membrane potential* [4, 24, 25]

$$u_{i,t} = \sum_{j \in \mathcal{P}_i} w_{ij}^e \big(\alpha_t * z_{j,t}\big) + w_i^e \big(\beta_t * z_{i,t-1}\big) + \bar{w}_i^e. \tag{15}$$

In (15), the contribution of each pre-synaptic neuron $j$ is given by the synaptic trace $\alpha_t * z_{j,t}$, where the sequence $\alpha_t$ is the synaptic *spike response* and $z_{j,t}$ is the pre-synaptic neuron's output, multiplied by the synaptic weight $w_{ij}^e$. The membrane potential is also affected by the post-synaptic trace $\beta_t * z_{i,t-1}$, where $\beta_t$ is the *feedback* response and $z_{i,t}$ is the post-synaptic neuron's output, which is weighted by $w_i^e$. The spike response $\alpha_t$ and feedback response $\beta_t$ can be different for visible and hidden neurons (see Appendix A.2), and they have memory limited for $\tau_e$ samples, i.e., $\alpha_t = \beta_t = 0$ for $t > \tau_e$. Finally, $\bar{w}_i^e$ is a learnable bias parameter. Choices for these functions and details on the neuron models are provided in Appendix A.2.

**Deterministic SRM.** For each neuron $i \in \mathcal{N}$, the binary output $z_{i,t} \in \{0, 1\}$ at time $t$ depends on its membrane potential $u_{i,t}$ as

$$z_{i,t} = \Theta(u_{i,t}), \tag{16}$$

where $\Theta(\cdot)$ is the Heaviside step function: A spike is emitted when the membrane potential $u_{i,t}$ is positive.

**SRM with stochastic threshold.** For each neuron $i \in \mathcal{Y}$ in the readout layer, the spiking probability, when conditioned on the overall spiking history, depends on the membrane potential as

$$p_{\boldsymbol{w}_i^e}(y_{i,t} = 1 | u_{i,t}) = \sigma(u_{i,t}), \tag{17}$$

where $\sigma(x) = 1/(1 + \exp(-x))$ is the sigmoid function, and $\boldsymbol{w}_i^e = \{w_{ij}^e, w_i^e, \bar{w}_i^e\}$ is the per-neuron set of parameters. This yields the encoding stochastic mapping (3) as

$$p_{\boldsymbol{w}^e}^{(\tau_e)}(\boldsymbol{y}||\boldsymbol{x}) = \prod_{i \in \mathcal{Y}} \prod_{t=1}^{T} p_{\boldsymbol{w}_i^e}(y_{i,t}||u_{i,t}). \tag{18}$$

**Gradients.** From Sec. 4, given (13)-(14), we need to compute the gradients $\nabla_{\boldsymbol{w}^e} \log p_{\boldsymbol{w}^e}^{(\tau_e)}(\boldsymbol{y}||\boldsymbol{x})$. For the readout neurons, this can be directly done as [4]

$$\nabla_{w_{ij}^e} \log p_{\boldsymbol{w}_i^e}(y_{i,t} = 1|u_{i,t}) = (\alpha_t * z_{j,t})(y_{i,t} - \sigma(u_{i,t})) = e_{ij,t}, \tag{19}$$

where we recall that $z_{j,t}$ is the spiking signal of pre-synaptic neuron $j$. In (19), we have defined the pre-synaptic *eligibility trace* $e_{ij,t}$. Learning rules can similarly derived for weights $w_i^e$ and $\bar{w}_i^e$, with corresponding eligibility traces $e_{i,t}$ and $\bar{e}_{i,t}$, and we define per-neuron eligibility trace $e_{i,t} = \{e_{ij,t}, e_{i,t}, \bar{e}_{i,t}\}$. For neurons $i \in \mathcal{H}$ in the hidden layer, we rely on e-prop [3] – an SG-based technique – to obtain the approximation

$$\nabla_{w_{ij}^e} \log p_{\boldsymbol{w}^e}^{(\tau_e)}(\boldsymbol{y}||\boldsymbol{x}) \approx \sum_{t=1}^{T} L_{i,t} e_{ij,t}, \tag{20}$$

with the learning signal

$$L_{i,t} = \sum_{k \in \mathcal{Y}} B_{ik}(y_{k,t} - \sigma(u_{k,t})) \tag{21}$$

characterized by random, but fixed, weights $B_{ik}$ following direct feedback alignment [26], and eligibility trace

$$e_{ij,t} = \Theta'(u_{i,t} - \vartheta)(\alpha_t * z_{j,t}). \tag{22}$$

The approximation (20) comes from: *(i)* truncating the BPTT by ignoring the dependence of the log-loss $-\log p_{\boldsymbol{w}_i^e}(y_{i,t}|u_{i,t})$ at time $t$ on the weight $w_i^e$ due to the previous time instants $t' < t$; *(ii)* replacing backpropagation through neurons, via feedback alignment; and *(iii)* using the derivative of a surrogate function $\Theta'(\cdot)$ [12]. For example, with a sigmoid surrogate gradient, we have $\Theta'(\cdot) = \sigma'(\cdot)$.

The overall algorithm, referred to as VDIB, is summarized in Appendix A.1. As anticipated in the previous sections, the updates (31) and (32) contain elements from both probabilistic and SG-based learning rules. In particular, the global learning signal $(\ell_{\boldsymbol{w}^d}(\boldsymbol{y}, \boldsymbol{r}) + \beta \cdot \ell_{\boldsymbol{w}^e}(\boldsymbol{y}, \boldsymbol{x}))$ relates to the global signal used in probabilistic learning [4, 24], whilst per-neuron error signal (21) is a feature of SG schemes [3, 12].

## 6 Experiments

We now evaluate the proposed VDIB solution in an experimental setting. We specifically focus on two tasks: predictive coding and naturalization of spiking signals. The former task highlights the aspect of causal, online, encoding and decoding, and relates closely with work in theoretical neuroscience [21]. In contrast, the latter task emphasizes the role of the proposed approach for neuromorphic computing [6]. Specifically, the naturalization of spiking inputs, e.g., obtained from a DVS camera [5], enables the integration of neuromorphic sensors with conventional digital sensors and processors [6].

As the proposed hybrid architecture gives flexibility in the choice of the decoding ANN, we have considered three options: (i) a simple logistic regression model, (ii) a multilayer perceptron (MLP) with one hidden layer, and (iii) a convolutional neural network (CNN) with two convolutional layers and one fully connected layer. As discussed in Sec. 3, at each time $t$, the decoding ANN is given the sequence $\boldsymbol{y}_{t-\tau_d}^t$ of $\tau_d$ time samples produced by the encoding SNN. As a benchmark, we also consider conventional **rate decoding**, whereby time samples within the sequence $\boldsymbol{y}_{t-\tau_d}^t$ are summed over time, yielding $\sum_{t'=t-\tau_d}^{t} \boldsymbol{y}_{t'}$, before being fed to the ANN.

By considering encoding networks with a layered architecture, our implementation of both ANN and SNN makes use of automatic differentiation and general-purpose deep learning libraries [27] and Tesla V100 GPUs.

### 6.1 Efficient predictive coding

In this first experiment, we explore the use of the proposed hybrid variational autoencoder for a predictive coding problem. Following the experiment proposed in [21], we consider exogenous signals consisting of 20 correlated spiking signals that represent two independent auto-regressive "drifting Gaussian blobs". At each time $t$, the position $p_t \in \{1, \ldots, 20\}$ of each "blob" across the exogenous inputs follows a (wrapped) Gaussian path $\mathcal{N}(\theta_t, \sigma)$, with $\theta_t$ evolving according to an

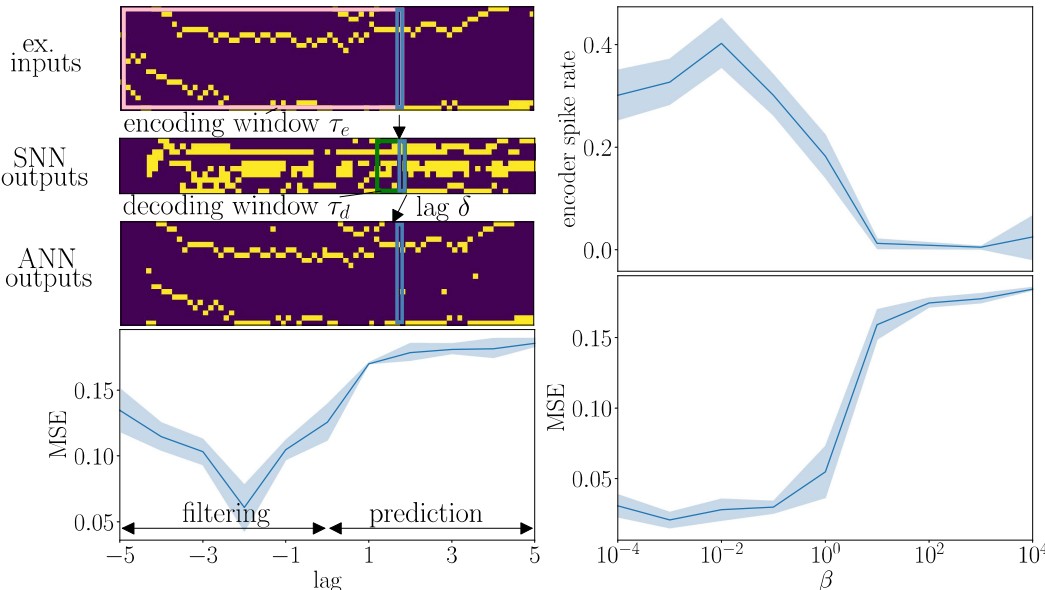

Figure 2: Left: Illustration of the exogenous inputs (top), and accuracy as a function of the target lag $\delta$ (bottom). Right: Encoding network spike rates with lag $\delta = -2$ versus regularization constant $\beta$ (top), and corresponding accuracy (bottom). Shaded areas represent standard deviations.

AR(2) process $\theta_t = \theta_{t-1} + v_t$, with $v_t = av_{t-1} - b\epsilon_t$, $\epsilon_t \sim \mathcal{N}(0, 1)$, $\sigma = 0.45$, $a = 0.9$ and $b = 0.14$. A realization of the exogenous inputs can be found in Fig. 2(left, top). The reference signal $\boldsymbol{r}_t$ at time-step $t$ consists of the one-hot encoding of the positions of the two blobs at time $t + \delta$ for some time lag $\delta \in \mathbb{Z}$. A one layer encoder compresses the 20 exogenous spiking signals into 10 spiking signals, which are decoded by a softmax regressor (playing the role of the decoding ANN). We choose the reference distribution $q(\boldsymbol{y})$ to be i.i.d. Bern(0.2) to enforce temporal sparsity. We set $\tau_e = 5$, and $\tau_d = 5$ and, where not mentioned otherwise, $\beta = 1$. Training is done over $50,000$ randomly generated examples, of length $T = 100$. Testing is done on a single example of length $T = 1,000$. Results are presented in terms of mean squared error (MSE), and averaged over 5 trials with random initializations. In the right panels, it can be seen that by modulating the value of $\beta$, one can explore the trade-off between temporal sparsity and accuracy. Overall, the approach is seen to yield compact spiking representations that enable the reconstruction of the input signal via online decoding. Finally, we provide a study of the effect of the window sizes $\tau_e$ and $\tau_d$ in Appendix A.3.

### 6.2 Naturalizing MNIST images

We now consider image naturalization. To start, we obtain exogenous input spiking signals $\boldsymbol{x}$ by encoding MNIST images using either Poisson encoding (see [25]) or time-to-first-spike encoding (as in [7]). The relevance signal is defined as $\boldsymbol{r}_t = \mathbf{0}_{N_X}$ for $t = 1, \ldots, T - 1$, and $\boldsymbol{r}_T$ is the original image. Results are provided in terms of the accuracy of the convolutional LeNet ANN trained on the original MNIST dataset [28]. Specifically, we feed the decoded natural images to the LeNet ANN, and report its accuracy. The hybrid variational autoencoder compresses the 784 exogenous spiking signals into 256 spiking signals at the readout layer of the SNN. We set $\tau_e = \tau_d = T = 30$, and $\beta = 0.001$. Training is carried over $200,000$ examples, and testing is done on the $10,000$ images from the test dataset. Other experimental details can be found in Appendix A.3.

Table 1: Comparison of classification accuracies for an MLP ANN decoder under various encoding strategies (for the MNIST digits at the input of the SNN) and decoding schemes (for the output of the SNN).

| Encoding \ Decoding | Rate | VDIB |
|---|---|---|
| Rate | $60.50 \pm 0.32\%$ | $\mathbf{91.32 \pm 0.27\%}$ |
| Time-to-First-Spike | $86.43 \pm 0.84\%$ | $\mathbf{92.82 \pm 0.25\%}$ |

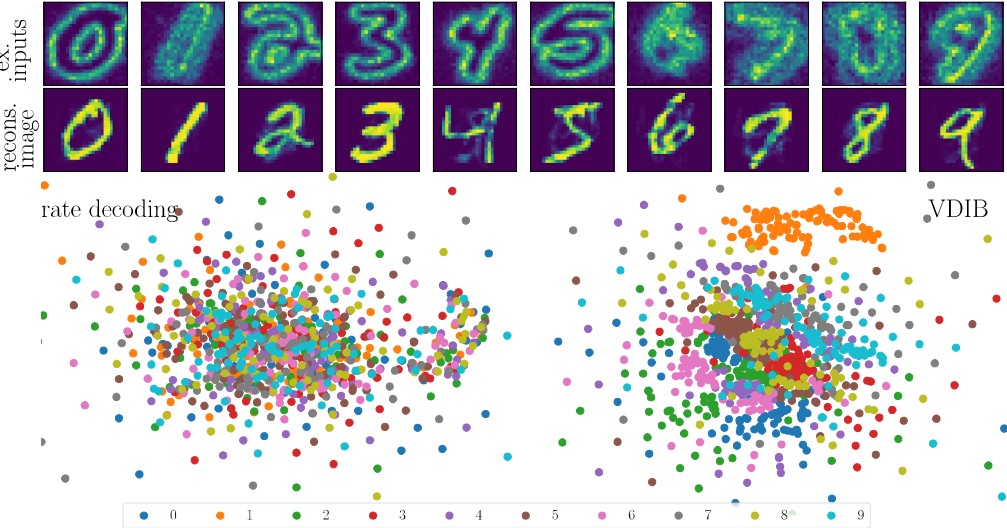

Figure 3: Top: Exogenous spiking inputs from the MNIST-DVS dataset summed over time samples (first row), and reconstructed natural images via VDIB (second row). Bottom: T-SNE representation of the spiking signals produced by the SNN with rate decoding (left), and with VDIB (right).

We explore the impact of various encoding and decoding strategies in Table 1, when using a decoding MLP. From Table 1, we note that, although rate decoding provides clear benefits in reducing the size of inputs to the decoder, this is at the cost of a large drop in accuracy. The clear performance advantages for time decoding validate our working hypothesis regarding the importance to move away from handcrafted decoding strategies.

### 6.3 Naturalizing MNIST-DVS images

Finally, we evaluate VDIB on the MNIST-DVS dataset [29]. Following [16, 30–32], we crop images to $26 \times 26$ pixels, and to 2 seconds. We also use a sampling period of 10 ms, which yields $T = 200$, and evaluate the impact of coarser sampling rates on the proposed method. To obtain natural images, the reference signal $r$ used during training is given by a single image of the corresponding class from the MNIST dataset. In this experiment, the 676 exogenous signals from MNIST-DVS images are compressed by the encoder into 256 spiking signals, which are fed to a convolutional decoder. We set $\tau_e = \tau_d = T$, and $\beta = 0.001$. Training is carried over $100,000$ examples, and testing is done on the $10,000$ images from the test dataset.

Fig. 3(top row) illustrates with some examples that VDIB can effectively naturalize MNIST-DVS images. In the bottom row, we show T-SNE [33] representations of encoded spiking signals produced by the SNN when using VDIB, and compare them with those produced when considering rate decoding. The latter is seen to be unable to produce representations from which class patterns can be identified, since all the representations are superimposed.

## 7 Conclusion

In this work, we have introduced VDIB, a variational learning algorithm for a hybrid SNN-ANN autoencoder that is based on a directed, causal, variant of the information bottleneck. The proposed system enables a shift from conventional handcrafted decoding rules to optimized learned decoding. VDIB can find applications in neuromorphic sensing through naturalization of spiking signals, and generally as a broad framework for the design of training algorithms for SNNs. This includes applications in mobile edge computing, offering opportunities in, e.g., personal healthcare. Limitations of this work include the choice of ANNs for decoders, where the usage of RNNs may result in improvements. Another line for future work consists in learning the reference distribution $q(\boldsymbol{y})$ instead of fixing it.

## Acknowledgments and Disclosure of Funding

This work was supported by the European Research Council (ERC) under the European Union's Horizon 2020 research and innovation programme (grant agreement No. 725731), by Intel Labs via the Intel Neuromorphic Research Community (INRC), and by a National Research Foundation of Korea (NRF) grant from the Government of Korea (MSIT) (No. 2021R1F1A1063288).

The authors acknowledge use of the Joint Academic Data science Endeavour (JADE) HPC Facility (http://www.jade.ac.uk/), and use of the research computing facility at King's College London, Rosalind (https://rosalind.kcl.ac.uk).

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
