# A Appendix

## A.1 Derivation of the bound

As described in Sec 4, using a decoding network defined by the causally conditional distribution $q_{\boldsymbol{w}^d}^{(\tau_d)}(\boldsymbol{r}||\boldsymbol{y})$ from eq. (7), we use a standard variational inequality [9] to lower bound the mutual information at each step $t = 1, \ldots, T$

$$I_{\boldsymbol{w}^e}^{(\tau_d)}(\boldsymbol{y}_{t-\tau_d}^t; \boldsymbol{r}_t|\boldsymbol{r}^{t-1}) \geq \mathrm{E}_{p_{\boldsymbol{w}^e}(\boldsymbol{y}^t, \boldsymbol{r}^t)}\left[\log \frac{q_{\boldsymbol{w}^d}(\boldsymbol{r}_t|\boldsymbol{r}^{t-1}, \boldsymbol{y}_{t-\tau_d}^t)}{p(\boldsymbol{r}_t|\boldsymbol{r}^{t-1})}\right] \tag{23}$$

$$= \mathrm{E}_{p_{\boldsymbol{w}^e}(\boldsymbol{y}^t, \boldsymbol{r}^t)}\left[\log q_{\boldsymbol{w}^d}(\boldsymbol{r}_t|\boldsymbol{r}^{t-1}, \boldsymbol{y}_{t-\tau_d}^t)\right] + H(\boldsymbol{r}_t|\boldsymbol{r}^{t-1}). \tag{24}$$

Summing over the time-steps, we then obtain

$$I_{\boldsymbol{w}^e}^{(\tau_d)}(\boldsymbol{y} \to \boldsymbol{r}) \geq \sum_{t=1}^{T} \mathrm{E}_{p_{\boldsymbol{w}^e}(\boldsymbol{y}^t, \boldsymbol{r}^t)}\left[\log q_{\boldsymbol{w}^d}(\boldsymbol{r}_t|\boldsymbol{r}^{t-1}, \boldsymbol{y}_{t-\tau_d}^t)\right] + \sum_{t=1}^{T} H(\boldsymbol{r}_t|\boldsymbol{r}^{t-1}) \tag{25}$$

$$= \mathrm{E}_{p_{\boldsymbol{w}^e}(\boldsymbol{y}, \boldsymbol{r})}\left[\sum_{t=1}^{T} \log q_{\boldsymbol{w}^d}(\boldsymbol{r}_t|\boldsymbol{r}^{t-1}, \boldsymbol{y}_{t-\tau_d}^t)\right] + H(\boldsymbol{r}) \tag{26}$$

$$= \mathrm{E}_{p_{\boldsymbol{w}^e}(\boldsymbol{y}, \boldsymbol{r})}\left[\log q_{\boldsymbol{w}^d}^{(\tau_d)}(\boldsymbol{r}||\boldsymbol{y})\right] + H(\boldsymbol{r}), \tag{27}$$

where the entropy of the target signals $H(\boldsymbol{r})$ is independent of the model parameters $\boldsymbol{w}^e$ and can be ignored for the purpose of training.

To bound the second information term in (6), we introduce an arbitrary auxiliary distribution $q(\boldsymbol{y}) = \prod_t q(\boldsymbol{y}_t|\boldsymbol{y}^{t-1})$ on the readout layer. As in the implementation in [9], we will consider it to be fixed, although it can potentially be optimized on. Using a standard variational inequality [9], we then obtain the upper bound the mutual information of the encoder for each $t = 1, \ldots, T$ as

$$I_{\boldsymbol{w}^e}(\boldsymbol{x}^t; \boldsymbol{y}^t) \leq \mathrm{E}_{p_{\boldsymbol{w}^e}(\boldsymbol{y}^t, \boldsymbol{x}^t)}\left[\log \frac{p_{\boldsymbol{w}^e}(\boldsymbol{y}_t|\boldsymbol{x}_{t-\tau_e}^t, \boldsymbol{y}^{t-1})}{q(\boldsymbol{y}_t|\boldsymbol{y}^{t-1})}\right]. \tag{28}$$

Summing over time-steps $t = 1, \ldots, T$, we obtain

$$I_{\boldsymbol{w}^e}(\boldsymbol{x} \to \boldsymbol{y}) \leq \mathrm{E}_{p_{\boldsymbol{w}^e}(\boldsymbol{y}, \boldsymbol{x})}\left[\log \frac{p_{\boldsymbol{w}^e}^{(\tau_e)}(\boldsymbol{y}||\boldsymbol{x})}{q(\boldsymbol{y})}\right]. \tag{29}$$

Combining bounds (25)-(29), we obtain the **variational DIB** loss $\mathcal{L}_{VDIB}(\boldsymbol{w}^e, \boldsymbol{w}^d)$ as an upper bound on the negative DIB objective

$$\mathcal{L}_{VDIB}(\boldsymbol{w}^e, \boldsymbol{w}^d) = \mathrm{E}_{p_{\boldsymbol{w}^e}(\boldsymbol{y}, \boldsymbol{r})}\left[-\log q_{\boldsymbol{w}^d}^{(\tau_d)}(\boldsymbol{r}||\boldsymbol{y})\right] + \beta \cdot \mathrm{E}_{p(\boldsymbol{x})}\left[\mathrm{KL}(p_{\boldsymbol{w}^e}^{(\tau_e)}(\boldsymbol{y}||\boldsymbol{x})||q(\boldsymbol{y}))\right]. \tag{30}$$

This completes the derivation. The proposed method is detailed in Algorithm 1.

## A.2 Neuron models

**Deterministic SRM.** For hidden neurons $i \in \mathcal{H}$, synaptic and feedback filters are chosen as the *alpha-function* spike response $\alpha_t = \exp(-t/\tau_{\mathrm{mem}}) - \exp(-t/\tau_{\mathrm{syn}})$ and the exponentially decaying feedback filter $\beta_t = -\exp(-t/\tau_{\mathrm{ref}})$ for $t \geq 1$ with some positive constants $\tau_{\mathrm{mem}}, \tau_{\mathrm{syn}}$, and $\tau_{\mathrm{ref}}$. It is then common to compute the membrane potential (15) using the following set of autoregressive equations [24]

$$p_{j,t} = \exp\left(-1/\tau_{\mathrm{mem}}\right)p_{j,t-1} + q_{j,t-1}, \tag{33a}$$

$$\text{with } q_{j,t} = \exp\left(-1/\tau_{\mathrm{syn}}\right)q_{j,t-1} + s_{j,t-1}, \tag{33b}$$

$$\text{and } r_{i,t} = \exp\left(-1/\tau_{\mathrm{ref}}\right)r_{i,t-1} + s_{i,t-1}. \tag{33c}$$

Equations (33a)-(33b) define the computation of the pre-synaptic contribution via a second-order autoregressive (AR) filter, while the feedback of neuron $i$ is computed in (33c) via a first-order AR filter.

**Algorithm 1:** Learn to Time-Decode via Variational Directed IB (VDIB)

**Input:** Exogeneous spiking signal $\mathbf{x}$, reference natural signal $\mathbf{r}$, reference distribution $q(\boldsymbol{y})$,
learning rate $\eta$, moving average parameter $\kappa$, regularization strength $\beta$
**Output:** Learned model parameters $\boldsymbol{w}^e, \boldsymbol{w}^d$

1 **initialize** parameters $\boldsymbol{w}^e, \boldsymbol{w}^d$;
2 **for** each iteration **do**
3      draw a sample $(\boldsymbol{x}, \boldsymbol{r})$ from the data set
4      **for** each time $t = 1, \ldots, T$ **do**
5          - generate spike outputs from the encoding SNN $\boldsymbol{y}_t \sim p_{\boldsymbol{w}^e}(\boldsymbol{y}_t \| \boldsymbol{x}^t)$
6          - a central processor collects the log-probabilities $\log p_{\boldsymbol{w}^e}(y_{i,t} \| \boldsymbol{x}^t)$ for all readout
          neurons $i \in \mathcal{Y}$ and updates the encoder loss

$$\ell_{\boldsymbol{w}^e}(\boldsymbol{y}_t, \boldsymbol{x}_t, \boldsymbol{h}_t) = \sum_{i \in \mathcal{Y}} \log \left( \frac{p_{\boldsymbol{w}^e}(y_{i,t} \| \boldsymbol{x}^t)}{q(y_{i,t})} \right)$$

7          - generate outputs from the decoding network $\boldsymbol{r}_t \sim q_{\boldsymbol{w}^d}(\boldsymbol{r}_t \| \boldsymbol{y}^t)$
8          - a central processor collects the log-probabilities $\log q_{\boldsymbol{w}^d}(\boldsymbol{r} \| \boldsymbol{y}^t)$ and updates the decoder
          loss

$$\ell_{\boldsymbol{w}^d}(\boldsymbol{y}_t, \boldsymbol{r}_t) = -\log q_{\boldsymbol{w}^d}(\boldsymbol{r}_t \| \boldsymbol{y}^t)$$

9          **for** each encoding network readout neuron $i \in \mathcal{Y}$ **do**
10             compute update $\Delta_{i,t}$ using (19) as

$$\Delta_{i,t} = \boldsymbol{e}_{i,t}$$

11          **end**
12          **for** each encoding network hidden neuron $i \in \mathcal{H}$ **do**
13             compute update $\Delta_{i,t}$ using (20) as

$$\Delta_{i,t} = L_{i,t} \cdot \boldsymbol{e}_{i,t}$$

14          **end**
15          update the encoding SNN model parameters as

$$\boldsymbol{w}^e \leftarrow \boldsymbol{w}^e - \eta \cdot \left( \ell_{\boldsymbol{w}^d}(\boldsymbol{y}_t, \boldsymbol{r}_t) + \beta \cdot \ell_{\boldsymbol{w}^e}(\boldsymbol{y}_t, \boldsymbol{x}_t, \boldsymbol{h}_t) \right) \cdot \Delta_t \tag{31}$$

16          update decoding ANN model parameters as

$$\boldsymbol{w}^d \leftarrow \boldsymbol{w}^d - \eta \cdot \nabla_{\boldsymbol{w}^d} \ell_{\boldsymbol{w}^d}(\boldsymbol{y}_t, \boldsymbol{r}_t) \tag{32}$$

17      **end**
18 **end**

---

**SRM with stochastic threshold.** For readout neurons, we associate each synapse with $K_a \geq 1$ spike responses $\{\alpha_t^k\}_{k=1}^{K_a}$ and corresponding weights $\{w_{ij,k}^e\}_{k=1}^{K_a}$, so that the contribution from pre-synaptic neuron $j$ to the membrane potential in (15) can be written as the double sum [34]

$$\sum_{j \in \mathcal{P}_i} \sum_{k=1}^{K_a} w_{ij,k}^e \left( \alpha_t^k * z_{j,t} \right). \tag{34}$$

By choosing the spike responses $\alpha_t^k$ to be sensitive to different synaptic delays, this parameterization allows SRM with stochastic threshold neurons to learn distinct temporal receptive fields [4, Fig. 5].

Table 2: Hyperparameters used in the different experiments.

| Parameter \ Experiment | Predictive coding | MNIST | MNIST-DVS |
|---|---|---|---|
| $\beta$ | 1 | $1e-3$ | $1e-3$ |
| $\eta$ | $1e-2$ | $1e-5$ | $1e-4$ |
| $N_X$ | 20 | 784 | 676 |
| $N_Y$ | 10 | 256 | 256 |
| $N_R$ | 210 | 784 | 784 |
| Decoder | Linear | MLP/Conv | Conv |
| Encoder architecture | $N_Y$ | $600 - N_Y$ | $800 - N_Y$ |
| $\tau_d$ | 5 | $T$ | $T$ |
| $\tau_e$ | 5 | $T$ | $T$ |
| $p$ | 0.2 | 0.3 | 0.3 |

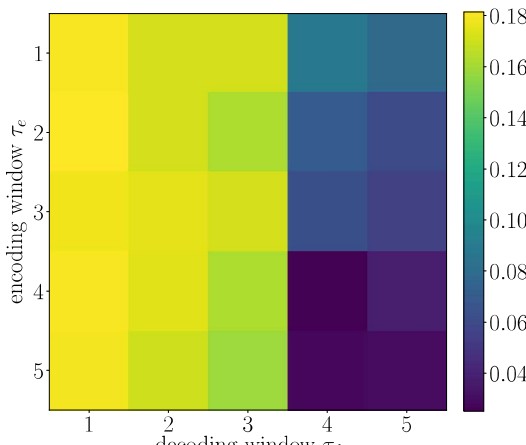

Figure 4: MSE for efficient filtering ($\delta = 3$) as a function of the window sizes $\tau_e$ and $\tau_d$.

### A.3 Experimental details

**Decoder architectures.** As explained in Sec. 6, we use three types of decoders: a fully-connected linear model, an MLP, and a convolutional ANN. The MLP comprises two layers, with architecture $(N_Y \times \tau_d)/2 - N_R$. The convolutional ANN has two convolutional layers with ReLU activation and a fully-connected layer, with architecture $\frac{T}{2}\texttt{c3p1s2} - 20\texttt{c3p1s2} - N_R$, where $X\texttt{c}Y\texttt{p}Z\texttt{s}S$ represents $X$ 1D convolution filters ($Y \times Y$) with padding $Z$ and stride $S$.

**Training hidden neurons with SG.** Part of the code used for feedback alignment has been adapted from `https://github.com/ChFrenkel/DirectRandomTargetProjection/`, distributed under the Apache v2.0 license.

**Efficient predictive coding.** In this experiment, we consider a logistic regression model for the decoder, i.e., $q_{\boldsymbol{w}^d}^{(\tau_d)}(\boldsymbol{r}_t|\boldsymbol{y}_{\tau_d}^t) = \mathcal{S}(\boldsymbol{r}_t|f_{\boldsymbol{w}^d}(\boldsymbol{y}_{\tau_d}^t))$ where $\mathcal{S}(\cdot)$ is the softmax function and $f_{\boldsymbol{w}^d}(\boldsymbol{x}) = \boldsymbol{w}^d\boldsymbol{x} + \boldsymbol{b}$ with learnable bias $\boldsymbol{b}$. We choose $N_X = N_Y = 20$, which gives $N_R = \binom{20}{1} + \binom{20}{2} = 210$. Other hyperparameters are precised in Table 2. Results are averaged over five trials with random initializations of the networks.

In Fig. 4, we analyze the effect of the window lengths $\tau_e$ and $\tau_d$ for filtering (lag $\delta = -3$). The MSE is seen to decrease as the window sizes $\tau_e$ and $\tau_d$ grow larger, which allows the SNN and ANN to access more information when estimating an input sample. The encoding window is seen to have a prominent effect, with a large improvement in MSE for $\tau_e > 3$, i.e., when the system is fed the signal to reconstruct.

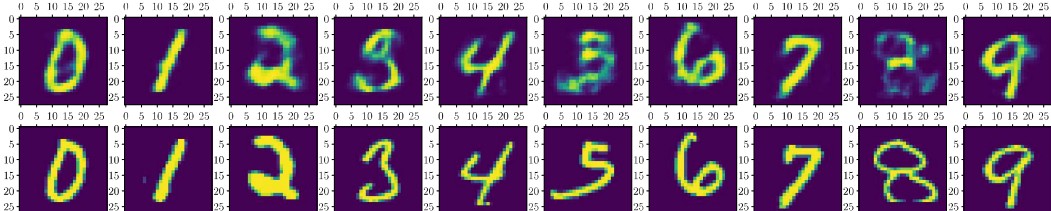

Figure 5: Top: Reconstructed MNIST digits. Bottom: Corresponding original target images.

Table 3: Comparison of classification accuracy with various encoding and decoding strategies for naturalization of MNIST digits, with MLP and convolutional ANN decoders.

| Encoding \ Decoding | Rate (MLP) | VDIB (MLP) | VDIB (Conv) |
|---|---|---|---|
| Rate | $60.50 \pm 0.32\%$ | $\mathbf{91.32 \pm 0.27}\%$ | $88.73 \pm 1.06\%$ |
| Time-to-First-Spike | $86.43 \pm 0.84\%$ | $\mathbf{92.82 \pm 0.25}\%$ | $90.72 \pm 0.39\%$ |

**MNIST.** The MNIST dataset is freely available online and has been obtained from `https://pytorch.org/vision/stable/_modules/torchvision/datasets/mnist.html`. It consists of $60,000$ training examples from ten classes, and $10,000$ test examples. Results are averaged over three trials with random initializations of the networks. In Fig. 5, we show examples of reconstructed digits with time encoding, and an MLP decoder. This validates that, as well as being able to be classified with high accuracy, the reconstructions look accurate. We also provide additional results on accuracy, using a convolutional decoder. This choice causes a small degradation in the performance of the system, but alleviates requirements on the GPU memory. Code and weights for the LeNet network have been obtained from `https://github.com/csinva/gan-vae-pretrained-pytorch` and are freely available.

Table 4: Comparison of the reconstruction MSE of MNIST-DVS digits for a convolutional ANN decoder under two decoding schemes (for the output of the SNN).

| Decoding | Rate | VDIB |
|---|---|---|
| MSE | $0.19 \pm 0.02$ | $\mathbf{0.02 \pm 0.0006}$ |

**MNIST-DVS.** The MNIST-DVS dataset is freely available online from `http://www2.imse-cnm.csic.es/caviar/MNISTDVS.html`. Examples have been obtained by displaying MNIST digits on a screen to a DVS camera. More information on the dataset can be found from [29].

In Table 4, we provide a quantitative measure of the results obtained for the reconstruction of MNIST-DVS digits. Specifically, we report the MSE between reconstructed and target MNIST digits with two decoding strategies. VDIB is shown to clearly outperform rate decoding.

**Reproducibility.** Code for this work can be found at `https://github.com/kclip`.