# OpenReview forum: "Learning to Time-Decode in Spiking Neural Networks Through the Information Bottleneck"
_NeurIPS.cc/2021/Conference — NeurIPS 2021 Poster_

### Official Review · Reviewer_Y1vr · 2021-06-25

**Rating:** 5
**Confidence:** 4

**Summary:**

Summary.\\
This work proposes a hybrid model for the naturalization and classification of spiking signals in an unsupervised manner. The proposed hybrid model consists of an SNN-based encoder and an ANN-based decoder and is constrained by the DIB, including the encoding and decoding discrepancies. This idea is interesting.

General Overview in My Mind.\\
Despite the interest of the overall goal of the research, this paper suffers from several limitations,  which make it improper for acceptance.

**Limitations And Societal Impact:**

Mentioned above.

**Main Review:**

Summary.

This work proposes a hybrid model for the naturalization and classification of spiking signals in an unsupervised manner. The proposed hybrid model consists of an SNN-based encoder and an ANN-based decoder and is constrained by the DIB, including the encoding and decoding discrepancies. This idea is interesting.

Readability.

The soundness and objective of the method, particularly its formalization and theoretical derivation, are hard to understand and follow.

[1] The use of some symbols is confusing, e.g., ";" in Eq. (2).

[2] As seen in Line 179, why can the unbiased estimation of gradients be achieved with only one sample?

[3] What's the binary output $h_{i,t}$?

[4] Figure 1 is hard to follow. Firstly, what's the pre-neuron learning signal $L_{i,t}$? Is it the same as $u_{i,t}$ in Eq. (15)? Secondly, what's the local eligibility trace $\boldsymbol{e}_{i,t}$? How to calculate it? Thirdly, it's better to mark the crucial symbols, such as $\boldsymbol{x}$, $\boldsymbol{r}$, and $\boldsymbol{r}$, on the roadmap of Figure 1(e). Finally, it's common sense that one can obtain the spiking inputs $\boldsymbol{x}_t$ from a static image using DVS or Poisson encoding, whereas how to reconstruct the picture, namely natural target signal, from the sequence $\boldsymbol{t}$?

Clarity and Quality.

[1] For each spiking neuron in the readout layer, the signals are generated by SRM with stochastic thresholds. So the symbols $\tau_e$ in Eq. (3) and $\tau_d$ in Eq. (4) are not fixed, right? Furthermore, how to ensure that there is an unbiased estimation of gradients in Eqs. (11) and (12)?

[2]  The experiments confused me since I lack some precision and information to conclude from the experiments. (1) How to evaluate the performance of the "naturalization" task? It's insufficient not to visualize some specific digits in Figure 3. (2) How to conclude the sentence "The latter is seen to be unable to produce representations from which class patterns can be identified." in Line 205?

[3] The comparative results in Table 1 are not promising since there lack of more contenders.

Minor Points.

[1]  There is a lot of blank space at the end of Page 6.

[2]  The format of the reference is also extremely irregular. For example, it is "Neural Computation" rather than "Neural computation" at first glance, and further, the formats of [19] and [20] are not consistent.

Anonymity.

Anonymity is not difficult to break since the authors uploaded an arxiv version of this manuscript on June 2nd, well after the deadline of NeurIPS 2021. I do not take this as an argument for rejection, but the authors should pay more attention to this.

General Overview in My Mind.

Despite the interest of the overall goal of the research, this paper suffers from several limitations,  which make it improper for acceptance.

**Time Spent Reviewing:**

2.5 hours

---

> ### Author Response · Authors · 2021-08-09
> **Authors response to reviewer Y1vr**
>
> Thank you for your detailed comments.
>
> Readability. \
> [1] The use of some symbols is confusing, e.g., ";" in Eq. (2). \
> (A) We have used standard notation, e.g., the use of the semi-colon is as in the reference textbook by Cover and Thomas, Elements of Information Theory (Wiley, 2001).
>
> [2] As seen in Line 179, why can the unbiased estimation of gradients be achieved with only one sample? \
> (A) Unbiasedness does not depend on the number of samples, as it is a property of the expectation of the estimator. You can refer to a standard textbook on estimation such as Kay’s Fundamentals of Statistical Processing, Volume I: Estimation Theory (1993, Pearson).
>
> [3] What's the binary output h? \
> (A) In the final version of the paper, we will avoid the notation $h_{i,t}$ which indeed was not correctly introduced.
>
> [4-1] Figure 1 is hard to follow. Firstly, what's the pre-neuron learning signal Li,t? Is it the same as ui,t  in Eq. (15)? Secondly, what's the  local eligibility trace ei,t? How to calculate it? Thirdly, it's better to mark the crucial symbols, such as x, r, and r, on the roadmap of Figure 1(e). \
> (A) The learning signal is defined in eq. (21), and the local eligibility trace in eq. (22). The discrepancy in the indexing of the eligibility trace between the figure and the text will be fixed. We will also modify the figure by including relevant quantities x, y, r, as you suggested.
>
> [4-2] Finally, it's common sense that one can obtain the spiking inputs from a static image using DVS or Poisson encoding, whereas how to reconstruct the picture, namely natural target signal, from the sequence ? \
> (A) In the literature, obtaining natural signals is done via handcrafted target spiking signals, which implicitly fixes the mechanisms used to decode spike, e.g., rate decoding for Poisson encoding. However, the arbitrary choice of target signals and decoding rule generally impairs the capacity of SNNs to encode and process information in the timing of spikes. Producing and leveraging relevant target signals for SNNs, and transforming them into natural signals is a challenging open problem, and is precisely the goal of this paper.
>
> Clarity and Quality. \
> [1] For each spiking neuron in the readout layer, the signals are generated by SRM with stochastic thresholds. So the symbols in Eq. (3) and in Eq. (4) are not fixed, right? Furthermore, how to ensure that there is an unbiased estimation of gradients in Eqs. (11) and (12)? \
> (A) As explained in the Notations paragraph of Sec. 3, the parameters $\tau$ are fixed as they represent the operation of encoder and decoder. This can also be also seen from the experiments. As mentioned above, unbiasedness does not depend on the number of samples, as is a property of the expectation of the estimator.
>
> [2-1] The experiments confused me since I lack some precision and information to conclude from the experiments. (1) How to evaluate the performance of the "naturalization" task? It's insufficient not to visualize some specific digits in Figure 3. \
> (A) Evaluating the performance of such tasks is an open and difficult problem. We have used two performance metrics, namely the accuracy of the decoding ANN as a quantitative measure, and the t-SNE visualization for a qualitative assessment. Since the ultimate goal of one of the experiments is to reconstruct natural images, we have also shown examples of reconstructions for the reader to evaluate. We believe that these are standard means to evaluate the performance of feature extraction models and the quality of synthesized images (see, e.g., [Stewart et al. ‘21], and [Isola et al. ‘18]). We will also add to the Supplementary Material the mean squared error between reconstruction and original images as a quantitative measure of the quality of the reconstruction for the example in Fig. 3.
>
> [2-2] How to conclude the sentence "The latter is seen to be unable to produce representations from which class patterns can be identified." in Line 205? \
> (A) We will clarify this point as “The latter is seen to be unable to produce representations from which class patterns can be identified, since all the representations are superimposed”.
>
> [3] The comparative results in Table 1 are not promising since there lack of more contenders. \
> (A) In our opinion, we have included all the relevant contenders in the table, and we would appreciate specific suggestions. Please note that we also have added a further "contender" in the Supplementary Material, namely VDIB with a Convolutional decoder.
>
> Minor Points. \
> [1] There is a lot of blank space at the end of Page 6. \
> (A) We will fix this.
>
> [2] The format of the reference is also extremely irregular. For example, it is "Neural Computation" rather than "Neural computation" at first glance, and further, the formats of [19] and [20] are not consistent. \
> (A) Thank you for noting this, we will fix any inconsistencies in the references.
>
> Anonymity. \
> [Q] Anonymity is not difficult to break since the authors uploaded an arxiv version of this manuscript on June 2nd, well after the deadline of NeurIPS 2021. I do not take this as an argument for rejection, but the authors should pay more attention to this. \
> (A) We have followed the conference guidelines as reported in https://neurips.cc/Conferences/2021/CallForPapers.

---

### Official Review · Reviewer_H28A · 2021-07-14

**Rating:** 6
**Confidence:** 4

**Summary:**

The authors proposed an end-to-end learning framework for learning spiking signal representations and corresponding decoding ANN. The framework optimizes a directed information bottleneck training criterion via SG and is demonstrated by experiments on various tasks on MNIST data. The contribution includes automatically generating spiking latent representations which are handcrafted before, generalizing standard information bottleneck principle to directed information bottleneck variation, developing the variational approximation of DIB objective, successfully applying the proposed framework on various tasks on MNIST.

**Ethical Concerns:**

No ethical concern was found.

**Limitations And Societal Impact:**

Limitations are briefly addressed in the conclusion section.
No negative social impacts are found.

**Main Review:**

minor problem:
  1. definition is missing for random vector a and b with only superscript (i.e. a^{t-1} := a^{t-1}_{0})
  2. some papers are cited twice in the same bracket.

  This paper focuses on spiking neural networks. Its SNN part is assembled with existing SNN works, and its ANN part is simple LR/MLP/CNNs. The main contribution of this paper is lying on adapting information bottleneck (IB) to the spiking signal encoding problem, and afterwards to train the SNN inside an autoencoder framework. The authors discover automatically efficient time-decoding rules from spiking signals produced by the encoding SNN. This claim somehow can be found in most existing papers on encoding-decoding architectures. The technical challenge itself is limited (for the DIB generalization from standard IB), although derivation steps are well written.

  The insight of applying IB in this problem is reasonable. The authors also claimed their proposed directed information bottleneck (DIB) objective is considered first time for spiking signal encoding. The difference w.r.t IB is to replace the mutual information with \tau-directed information. Generalized from directed information concept, \tau-directed information is designed to improve the flexibility of quantifying casual statistical dependencies within a limited window. Since the authors mentioned a lot of causality, we hope there could be more discussion on how to tune \tau_d and \tau_e in SNN and ANN part, and what could be inferred or observed from these critical parameters.

  This paper is relatively good but there is still space to improve. If the authors are trying to look deeper in the DIB aspect, targeting tasks compatible with this framework could be pretty much expanded so that you do not have to only play with MNIST data. If focusing on the SNN, it will be beneficial to give more detailed interpretations of your model and effects in the experiment, as what the cited paper [21, Matthew Chalk et al. PNAS 2018] does.

**Time Spent Reviewing:**

5

---

> ### Author Response · Authors · 2021-08-09
> **Authors response to reviewer H28A**
>
> (Q) Definition is missing for random vector a and b with only superscript (i.e., a^{t-1} := a^{t-1}\_{0}) \
> (A) We will clarify the notation $a^{t}$, which corresponds to $a_{1}^{t}$ as defined in the manuscript.
>
> (Q) Some papers are cited twice in the same bracket. \
> (A) We will also fix discrepancies in the references and typos in the text.
>
> (Q) Since the authors mentioned a lot of causality, we hope there could be more discussion on how to tune tau_d and tau_e in SNN and ANN part, and what could be inferred or observed from these critical parameters. \
> (A) In the final version, we will include in the Supplementary Material a further analysis of the impact of $\tau_d$ and $\tau_e$ in the context of predictive and efficient coding. More specifically, we will provide a table showing how the MSE evolves as a function of both $\tau_e$ and $\tau_d$ for the task presented in our document in two situations: filtering (negative lag delta), and prediction (positive lag delta).

---

### Official Review · Reviewer_9EcL · 2021-07-16

**Rating:** 8
**Confidence:** 2

**Summary:**

Proposes a way to effectively define spike codings for target labels, to optimize (or at least not inadvertently cripple) an SNN's ability to train and classify.
This combines a novel training regime applied to an SSN (as front end) with an ANN (as back end) in a VAE-type structure.

**Ethical Concerns:**

N.A.

**Limitations And Societal Impact:**

N.A.

**Main Review:**

This is an interesting, original and well-written paper, with potentially high value, to both neuroscience applications (spike coding) and ML applications (opening up the option of using SNNs for ML tasks).

Limitation: While I have confidence in the value of the sections that I could follow, sections 4 and 5 blew over my head without even ruffling my hair (too technical). So I have no opinion about those sections, and I'm happy to defer final assessment of the paper to better informed reviewers.

Misc comments:
Figures: excellent, especially the comprehensive caption.

62: SNNs (plural)

68, 69: introduce acronyms eg SG, BPTT

eqn 1: Notation difficulty: a^{t-1} is not defined and is unclear. I'm guessing the superscript should be a subscript; or is it missing a subscript? Is the intent to condition on tau samples of b and also on the previous sample of a?
eqn 2: same issue, with b^{t-1}, also eqn 3, 7, etc

103, 185: typos

136 - 137, eqn 5: Reason/motivation for this?

Table 1: The content was unclear to me on first examination. A more informative caption would help, and a clearer top left entry. "time" -> "time-to-first spike". Also, moving Table 1 to later in the paper (after it is mentioned) would smooth things for the reader.

297: It sounds like r was not the corresponding MNIST sample, but a different sample from the same digit class. Why this choice for r?

**Time Spent Reviewing:**

2

---

> ### Author Response · Authors · 2021-08-09
> **Authors response to reviewer 9EcL**
>
> Thank you for your thoughtful comments.
>
> (Q) 68, 69: Introduce acronyms, e.g., SG, BPTT \
> (A) Please note that acronyms are defined in the introduction.
>
> (Q) Eqn 1: Notation difficulty: a^{t-1} is not defined and is unclear. Eqn 2: Same issue, with b^{t-1}, also Eqn 3, 7. \
> (A) We will clarify the notation $a^{t}$, which corresponds to $a_{1}^{t}$ as defined in the manuscript.
>
> (Q) 103, 185: typos \
> (A) We will fix these.
>
> (Q) 136-137: Eqn 5: Reason/motivation for this? \
> (A) Eq. (5) integrates, using the chain rule of probability, the unknown population distribution and the SNN encoder mapping. We will clarify this point in the revised paper.
>
> (Q) Table 1: The content was unclear to me on first examination. A more informative caption would help, and a clearer top left entry. “time” -> “time-to-first-spike”. Also, moving Table 1 to later in the paper (after it is mentioned) would smooth things for the reader. \
> (A) In the final version of the paper, we will clarify the caption in Table 1 as “Comparison of classification accuracy using an MLP ANN decoder under various encoding strategies (for the MNIST digits at the input of the SNN) and decoding schemes (for the output of the SNN)”. We will also modify the left entry as you suggested, and place it later in the text.
>
> (Q) 297: It sounds like r was not the corresponding MNIST sample, but a different sample from the same digit class. Why this choice for r? \
> (A) There is no clear correspondence between individual MNIST-DVS and MNIST examples, which prompted our choice of choosing an arbitrary prototypical MNIST image as the reference signal.  In this regard, based on some preliminary experiments (not reported), we hypothesize that enhancing the diversity of reference targets for different examples generally improves the quality of the representation. In light of this, we view our results showing that VDIB creates high-quality reconstructions despite having access only to a single target per class as a sign of the robustness and effectiveness of the system.

---

### Official Review · Reviewer_Hpuu · 2021-07-16

**Rating:** 7
**Confidence:** 3

**Summary:**

In the paper "Learning to Time-Decode in Spiking Neural Networks Through the Information Bottleneck" the author propose a hybrid variational autoencoder architecture with an encoding spiking neural network and a decoding artificial neural network. The authors evaluate the method on predictive coding and naturalization of spiking signals tasks.

**Limitations And Societal Impact:**

Limitations with respect to choice of decoders and the reference distribution are very briefly mentioned in the Conclusion section.

**Main Review:**

Originality:
The authors devise a novel hybrid architecture that lets an ANN learn how to decode spikes and train it together with an SNN encoder, thereby not fixing the coding a priori. I have not seen this idea before and find it intriguing. To this end, the authors introduce a novel directed information bottleneck objective. Overall, the original contributions of this work are excellent.

Quality:
The quality of the work is outstanding. The derivations of the variational directed information bottleneck algorithm and associated learning rules are rigorous. The predictive coding and naturalization tasks are sensible and demonstrate well the capabilities of the method.
A minor point: in lines 40-41 the authors confused panels a and b.

Clarity:
The manuscript is very well written. Figures can be understood relatively easily. There is a notations section which is helpful.

Significance:
The authors mention important practical applications in neuromorphic sensing. I believe learning optimal decoding strategies for SNN encoders could also lead to new predictions in systems neuroscience.

**Time Spent Reviewing:**

4

---

> ### Author Response · Authors · 2021-08-09
> **Authors response to reviewer Hpuu**
>
> Thank you for your positive review. We will fix the point you noted.

---

### Decision · Program_Chairs · 2021-09-27

**Decision:**

Accept (Poster)

**Comment:**

I want to thank the authors and reviewers for engaging with this paper, as well as the reviewers for participating in an internal discussion.

We have some disagreement between the reviewers with several positive reviews and one leaning rejection.

After reading the paper, and all of the related discussion I'm going to recommend this paper for acceptance.  I feel as though work has sufficient Quality, Originality, Clarity and Significance for acceptance.

I believe that many of the remaining issues reviewers have are largely stylistic and are not enough to block acceptance.